# `ViLCo-Bench`: **VIdeo Language COntinual learning Benchmark**

**Tianqi Tang**[1] [*]
School of Computer Science Engineering,
University of New South Wales, Australia
`tianqi.tang@unsw.edu.au`

**Shohreh Deldari** [*]
School of Computer Science Engineering
University of New South Wales, Australia
`s.deldari@unsw.edu.au`

**Hao Xue**
School of Computer Science Engineering
University of New South Wales, Australia
`hao.xue1@unsw.edu.au`

**Celso De Melo**
DEVCOM Army Research Laboratory
USA
`celso.miguel.de.melo@gmail.com`

**Flora Salim**
School of Computer Science Engineering,
University of New South Wales, Australia
`flora.salim@unsw.edu.au`

## Abstract

Video language continual learning involves continuously adapting to information from video and text inputs, enhancing a model's ability to handle new tasks while retaining prior knowledge. This field is a relatively under-explored area, and establishing appropriate datasets is crucial for facilitating communication and research in this field. In this study, we present the first dedicated benchmark, `ViLCo-Bench`, designed to evaluate continual learning models across a range of video-text tasks. The dataset comprises ten-minute-long videos and corresponding language queries collected from publicly available datasets. Additionally, we introduce a novel memory-efficient framework that incorporates self-supervised learning and mimics long-term and short-term memory effects. This framework addresses challenges including memory complexity from long video clips, natural language complexity from open queries, and text-video misalignment. We posit that `ViLCo-Bench`, with greater complexity compared to existing continual learning benchmarks, would serve as a critical tool for exploring the video-language domain, extending beyond conventional class-incremental tasks, and addressing complex and limited annotation issues. The curated data, evaluations, and our novel method are available at `https://github.com/cruiseresearchgroup/ViLCo`.

## 1 Introduction

Recent advancements in large multimodal foundation models have significantly revolutionized the field of multimodal machine learning. These models are typically trained on static large-scale datasets, which can be accessed simultaneously. However, real-world applications, such as autonomous driving and household robots, require continuously incoming data streams and exhibit inevitable shifts in domain and data distribution. Re-training from scratch for new data is impractical due to the significant computational resources and time required. Therefore, adopting a Continual Learning

---

[*]Co-first authors

38th Conference on Neural Information Processing Systems (NeurIPS 2024) Track on Datasets and Benchmarks.
Approved for public release: distribution is unlimited.

(CL) approach that builds upon previous knowledge is essential for lifelong AI, leading to substantial research focused on developing machine learning methods adaptable to dynamic environments, such as non-independent and identically distributed (non-i.i.d.) data, emerging tasks, and novel classes.

However, existing CL methods are mostly designed for a single modality of data—be it image [27, 31], text [4], audio [49], or video [46]—without considering multiple data modalities and the variety of tasks they entail. Some recent work has explored multimodal data, but only for still-image and textual data [12, 46]. With the rise of embodied AI devices and the abundance of sensor data, there is an urgent need for multimodal ML models to learn collaboratively from diverse data sources. This is crucial for empowering embodied AI agents with natural language understanding while mastering other modalities, such as platforms [20, 28] for human-centric question-answering from videos. Current continual learning benchmarks, focusing mainly on images, videos, or text-image combinations with clean category annotations, may not effectively evaluate continual learning in a video-language setting. Thus, our initial step is to establish a new benchmark in video-language settings, encompassing more challenging multimodal tasks (non-classification tasks).

**Challenges:** To achieve this, several challenges need to be addressed: (1) Continuously generated data requires resources for annotation at the same pace, making access to quality labeled data daunting in CL

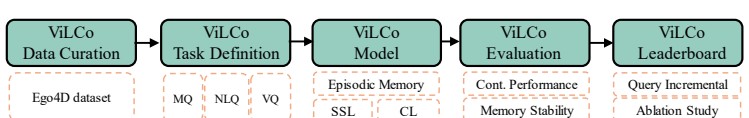

Figure 1: `ViLCo-Bench` benchmark pipeline.

setups. This highlights the necessity for self-supervised approaches. (2) Self-supervised learning (SSL) models themselves require adaptation for dynamic input data. Developing online/continual SSL models to address this challenge is an active research area, even in still image processing [6, 10, 15, 43]. [31] reviews the challenges of continual SSL models for video streaming applications. (3) Multimodal data, especially in video, can have queries based on various modalities (e.g., text, image, or time), resulting in diverse tasks beyond traditional classification. (4) The capacity to process and remember long videos poses a significant challenge for current video-based CL models. This highlights not only computational limitations but also memory issues in replay-based approaches relying on previously seen samples.

**Our Contributions:** Within the realm of multimodal continual learning (CL) tasks, specifically in the video-language domain, there is a notable absence of a unified definition for video-text CL tasks, hindering research progress. Contemporary approaches predominantly emphasize class-incremental CL, which is foundational in the uni-modal domain [46, 36, 45] but they do not address the unique challenges of video-language CL tasks [54]. Thus, there is a pressing need for new challenges and benchmarks tailored to the multimodal domain. Despite recent advances in multimodal CL, they primarily focus on text and still images. Existing models for continual learning in streaming data [36] are also limited. Therefore, we propose the creation of a video-text continual learning benchmark to provide a standardized platform for evaluating models in the challenging text-video context.

As depicted in Fig 1, `ViLCo-Bench` addresses three key problem statements: (1) *Definition of Video-Text Continual Tasks*: Establish clear definitions for video-text continual learning tasks, focusing on non-classification tasks such as episodic memory recall, cross-modal understanding, and multi-task learning in a continual setting. (2) *Setup of the Continual Learning Benchmark*: Define protocols for setting up the new CL benchmark, including considerations for train/validation splits to ensure fair and consistent assessment of models across various tasks. (3) *Limited labeled video-text samples*: We propose using a self-supervised technique with a novel use-case to mitigate the limited availability of video-text pairs and address variabilities in text descriptions. By establishing the first benchmark for video-text continual learning, we aim to foster innovation in multimodal continual learning.

The main contributions are: 1. We propose the first standardized benchmark in multimodal continual learning for video data, defining protocols for training and metrics for evaluation. This standardized framework allows researchers to effectively compare models, driving advancements in AI systems that can continuously learn from diverse data sources. 2. We define the setup for three recent multimodal tasks in a continual learning setup: Moment Query (MQ), Natural Language Query (NLQ), and Visual Query (VQ). We provide systematic insights into the challenges, gaps, and limitations of each video-text CL tasks. 3. We provide a comparison against four state-of-the-art (SOTA) models in

Table 1: Existing continual learning benchmarks

| CL Benchmark | data type | tasks | novelty |
|---|---|---|---|
| Core50 [27] | Image | classification/segmentation | New Instance/Class/Both |
| CLeAR [26] | Image | classification | non-IID stream of Images |
| CLiMB [42] | Image, Text | vision-text tasks | Language-based queries |
| vCLIMB [46] | Video | classification | Enhanced memory adaptation |
| TIC-DataComp [12] | Image, Text | classification, Retrieval | considering time-evolving data |
| ViLCo-Bench (Ours) | Video-Text | vision, text, moment tasks | New language-enabled tasks |

video and text continual learning for each benchmark setup. We prepared a curated dataset suitable for multimodal continual learning tasks using the well-known Ego4D dataset.

## 2   Backgrounds and Related Works

Recently, different benchmarks have been introduced for continual learning purposes in different tasks and modalities. CoRE50 [27], Stream-51 [37], and CLeAR [26] are among the most well-known benchmarks for continuous object recognition and streaming classification from still images. vCLIMB is the first that provides a standard benchmark for video continual learning [46]. The vCLIMB benchmark also proposes a novel approach that resolves the complexity of storing videos in memory by employing a temporal consistency loss, which reduces the number of frames stored. However, it focuses on class-incremental classification tasks and does not consider multimodal tasks.

On the other side, [55] provides a visual-question-answering (VQA) benchmark that includes image and text, but it still lacks video data and has not been covered for CL purposes. With recent advancements in multimodal foundation models and the capability of generative models to attend to different modalities (e.g., text, image, audio, video, code, etc.), the application of continual learning in multimodal data is becoming more diverse and inevitable. Despite the importance of the topic, there are no such benchmarks available yet, hence this will be the focus of this work. Table 1 summarizes existing benchmarks.

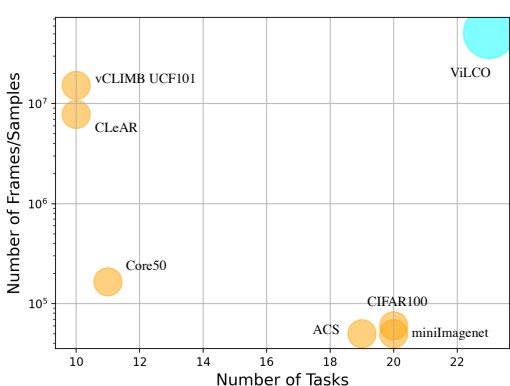

Figure 2: Comparisons with existing continual learning benchmarks.

Recent foundation models aim to learn transferable knowledge shared among modalities (e.g., video and corresponding text) that are generalizable to many downstream tasks [13, 7, 32]. Among these pre-training models, EgoVLP [25, 30] proposed the first comprehensive vision-language dataset using publicly available egocentric video dataset (Ego4D [16]) and formulated applicable tasks for vision-language processing such as Moment Query and Natural Language Query, along with vision only tasks such as instance retrieval and object detection tasks. However, they did not consider continual learning setups in their benchmark. More reviews on existing CL methods are provided in Appendix A.

## 3   ViLCo-Bench: Video-Language Continual Learning Benchmark

Unlike traditional continual learning tasks, ViLCo-Bench goes beyond classification and introduces challenges of cross-modal inference and temporal complexity in videos. We present three unique video-language continual learning tasks to evaluate existing methods and provide a curated dataset with annotations for each episodic memory task. Additionally, we propose novel memory-efficient models tailored for ViLCo-Bench.

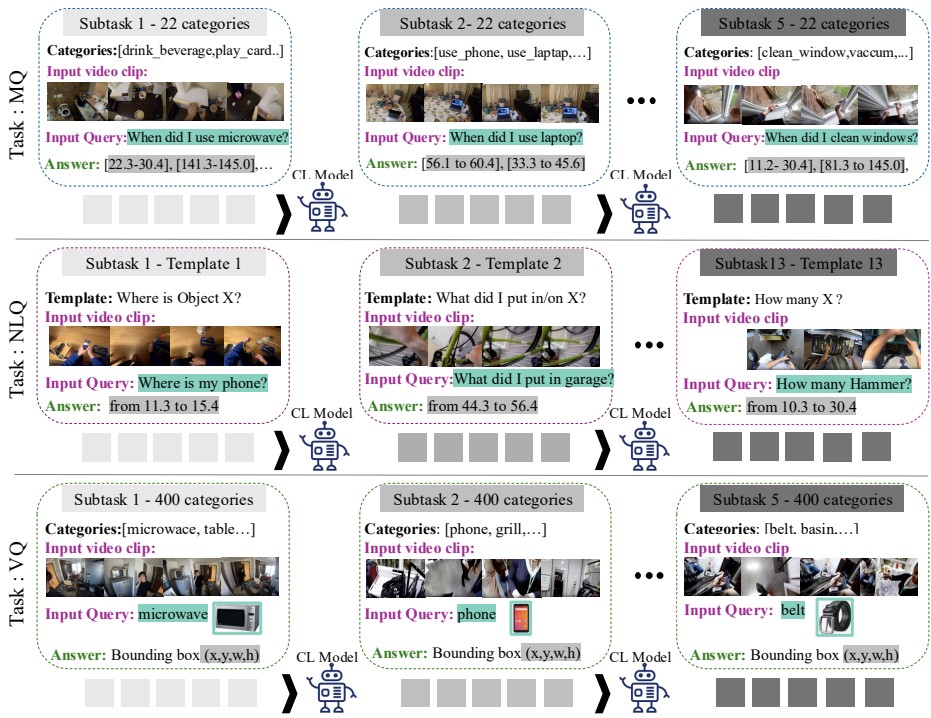

Figure 3: Illustration of the proposed query-incremental challenges.

## 3.1 Unique Video-Language Continual Learning Tasks

`ViLCo-Bench` proposes a novel continual learning setting, based on the largest egocentric video-language dataset Ego4D [16] specifically focusing on episodic memory tasks. Episodic memory refers to the ability to remember specific experiences and make past videos queriable [33], such as *"what did I eat"* and *"who did I sit by on my first flight to France?"*, which are different from semantic memory tasks like *"What's the capital of France?"*. This distinction becomes crucial in multimodal continual learning tasks, where models must retain and recall specific episodes in the past. In contrast to existing continual learning benchmarks, `ViLCo-Bench` presents three unique challenges. (1) **Exploring the Video-Language Domain**: `ViLCo-Bench` is the first benchmark to delve into the video-language domain in continual learning. The videos in this benchmark are significantly longer than those in previous video-based benchmarks. The average video length in our benchmark is 10 minutes, which is 85 times longer than the average video length in UCF101 [41]. (2) **Beyond Classification Tasks**: traditional continual learning benchmarks primarily focus on classification tasks. However, `ViLCo-Bench` introduces novel video-language tasks that require models to understand the semantic relationships between language queries and video content. (3) **Complex Annotation Splitting**: The annotation in video-language tasks (i.e., extracting temporal windows that each object/activity has appeared on the screen), is challenging. Moreover, there is overlap across labels which changes the problem to a multi-label multimodal learning problem. To address the aforementioned challenges, we introduce a novel setting called Query-Incremental Learning (QIL). The QIL includes three task formulations, each introducing unique complexities and necessitating specific model adaptations. As illustrated in Fig. 3, these tasks include:

**Moments Queries (MQ)**: For each MQ sub-task, the model identifies and recalls temporal windows for queries about specific human activities, such as "When did I drink a beverage?" or "When did I use the phone?" We split tasks based on action categories within the query space.

**Natural Language Queries (NLQ)**: In NLQ tasks, the model interprets and responds to queries about specific episodic memories. For example, given "What restaurant did I visit last Saturday?" the model must retrieve and identify the relevant temporal window.

**Visual Queries (VQ)**: VQ tasks involve processing visual information tied to episodic memories. For example, given a video of a garage workshop, the model should recall specific details like the

location of a toolbox when asked, "Where is the toolbox when I repair the bicycle?" VQ tasks are divided based on object categories.

## 3.2 Dataset Curation

We used Ego4D dataset [16] consisting of 3670 hours of egocentric videos . The tasks are to predict temporal windows or 2-dimensional bounding boxes from various queries mentioned before. These multimodal tasks introduce new challenges compared to unimodal (vision-only) models such as noisy videos due to quick head movements and limited field of view, free-form queries in natural language, and tiny response windows within lengthy video footage [33]. Therefore, we select videos and their corresponding queries and narrations to create three subsets for continual learning: `ViLCo-Bench-MQ`, `ViLCo-Bench-NLQ`, and `ViLCo-Bench-VQ`.

In `ViLCo-Bench-MQ`, we sample 165 hours of videos consisting of 110 action categories that include over $10,000$ videos varying between 8 to 10 minutes. We divide the task into five sub-tasks, each containing 22 action category queries. Annotations for each video may involve multiple classes and timestamps. `ViLCo-Bench-NLQ` consists of 136 hours of video. However, `ViLCo-Bench-NLQ` is more challenging because this subset leverages natural language as queries, such as "Where is Object X?", where "X" could be arbitrary textual descriptions. We summarize 13 query templates resulting in 13 sub-tasks. Query templates for NLQ scenarios are provided in Appendix B. `ViLCo-Bench-VQ` consists of 167 hours of videos based on the object categories from image queries. we divided it into 5 splits each containing over 400 categories.

To avoid the label overlapping issue across tasks and samples [40], we adopted a partitioning strategy to create and split training sets for different tasks and sub-tasks. Since over 90% of videos contain multiple classes spanning different subsets, we ensured each video was assigned to only one subset. We did this by prioritizing higher-frequency queries and excluding those that appeared in multiple subsets, thereby minimizing overlap and enhancing the clarity of learning objectives. This approach supports more effective training and evaluation of models in continual learning scenarios.

## 3.3 `ViLCo` Model

We investigate the impact of factors developed by existing continual learning methods [22, 1] within the context of video-language continual learning setting. Most of these techniques are developed for uni-modal benchmarks and focus on classification tasks. `ViLCo-Bench` introduces long-term videos and flexible natural language descriptions across various multimodal tasks. To align the video and language modalities within multimodal CL, we propose a memory-efficient framework by adapting a self-supervised learning pre-training mechanism and incorporating short-term and long-term memory modules (replay buffer). Fig 4 shows the overall architecture of `ViLCo-Bench` which consists of four key modules as introduced below:

**Video and Text Encoder.** Regarding the video data, in NLQ and MQ tasks, we adopt EgoVLPv2 [30] as the visual encoder. For VQ tasks, we used pre-trained ViT visual encoder by DINO [53]. We follow the same frame sampling strategy as EgoVLPv2. Moreover, we employ a CLIP (ViT-L/14) [32] to extract text embeddings across all types of tasks. In MQ tasks with multiple action categories per query, we concatenate them together.

**Cross-modal Encoder.** To fuse the video and language features we employ a Transformer network. Inspired by [24, 39], we also adopt a feature pyramid network to improve spatial-temporal representations by extracting features at multiple levels. This helps to understand temporal information in fine detail and precise localization prediction.

**Task-specific Head.** Given the variations in targets and annotations across different tasks, such as NLQ, we have implemented task-specific heads to address these distinctions. For MQ and NLQ tasks, we use a classification head and a localization head, respectively, to generate dense temporal predictions. It is important to note that in NLQ, the number of classes is typically one, as the queried events generally occur only once within an entire video. For VQ, our model employs a head designed to estimate the probability of the anchor-level queried object and a regression head to precisely adjust the coordinates of the bounding boxes.

**Episodic Memory Module.** To address long-term video analysis and multimodal interactions, it requires a more effective memory system. Existing CL models mostly rely on rehearsal (short-term)

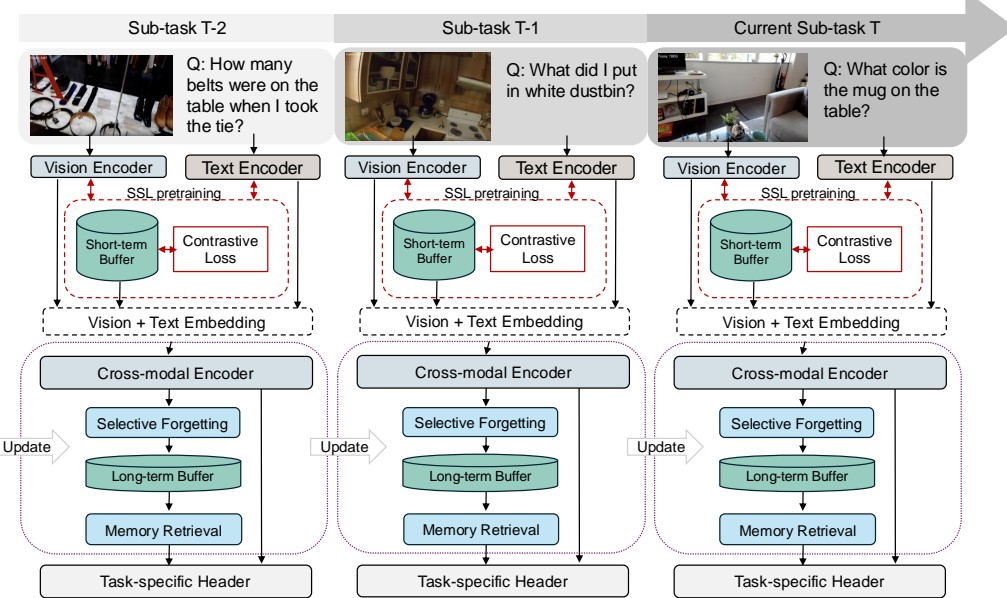

Figure 4: Overview of `ViLCo-Bench` employing episodic memory modules. All the modules except task-specific headers are shared among tasks and continually get updated.

buffers, which store a subset of seen examples from previous tasks [52, 17, 35]. These stored examples are periodically rehearsed or re-introduced into the learning process when new data is being learned. One of the updating strategies is to randomly select a set of examples from the incoming data stream, either replacing older entries or filling in available memory space [46]. In video language continual learning, maintaining most frame-level features in rehearsal buffers occupies substantial storage space, making it inefficient for long-term episode analysis and multimodal learning [34]. Thus, we propose a dual-memory module consisting of short-term and long-term replay buffers to dynamically facilitate episodic memory recall across varied events and tasks, as shown in Fig. 4. Both memory modules are initialized using the standard normal distribution.

Short-term memory component stores the text and vision embeddings across the current task. We extract negative samples from short-memory to be used for self-supervised training of the vision and text encoders. The primary objective of the long-term replay buffer is to preserve historical knowledge across similar queries from distinct tasks, thereby mitigating the risks of significant memory degradation. The buffer is characterized by a compact set of learnable parameters $\{(\mathcal{P}_1, \mathcal{K}_1), (\mathcal{P}_2, \mathcal{K}_2), ..., (\mathcal{P}_m, \mathcal{K}_m)\}$ where $\mathcal{P}$ represents the encoded prompt for episodic memory. $\mathcal{K}$ is a learnable key associated with value $\mathcal{P}$, dimensioned at $L \times D$. $L$ denotes the length of inputs, and $D$ indicates the embedding size, while $m$ represents the total memory capacity. We would like to automatically select the prompt by matching the language query and key matrix. To this end, we calculate the similarity between query embeddings and prompt keys, selecting the top-N prompts that best represent episodic memory. These selected prompt embeddings are then used to replace the original query embeddings, making the framework task-agnostic and revisiting analogous knowledge across different tasks. At each training step, we update $K$ and $P$ by minimizing the end-to-end training loss function. We employ a contrastive loss to regulate and update the prompt pool. This loss helps select prompts that minimize the distance between the task embeddings and the relevant keys, while maximizing the distance from irrelevant prompts (See Appendix **??**).

**Self-supervised Learning with Narrations.** We use SSL to enhance cross-modal representations by incorporating augmented video narrations. The challenge of collecting annotations for video-language tasks limits model generalization. Inspired by self-supervised methods [32, 11], we leverage additional video narrations from the Ego4d dataset to reduce video-language misalignment. By adopting contrastive learning, we learn robust representations from these narrations. To prevent forgetting in CL, we sample and store a few narration examples from previous tasks to construct negative pairs in short-term memory.(See Appendix C).

Table 2: Results of the state-of-the-art and our method on Moment Query.

| Method | Num. Task | Mem. Capacity | BwF↓ | Avg R@1 (%)↑ | | | Avg R@5 (%)↑ | | |
|---|---|---|---|---|---|---|---|---|---|
| | | | | IoU=0.3 | IoU=0.5 | mean | IoU=0.3 | IoU=0.5 | mean |
| Upper-Bound | None | None | None | $48.07_{\pm 0.09}$ | $38.71_{\pm 0.02}$ | 43.39 | $67.30_{\pm 0.03}$ | $56.87_{\pm 0.005}$ | 62.09 |
| Lower-Bound | None | None | None | $19.62_{\pm 0.25}$ | $10.87_{\pm 0.06}$ | 15.25 | $31.61_{\pm 0.76}$ | $19.11_{\pm 0.41}$ | 25.36 |
| EWC [22] | 5 | None | $24.2_{\pm 0.03}$ | $17.61_{\pm 0.57}$ | $12.51_{\pm 0.14}$ | 15.06 | $28.13_{\pm 0.03}$ | $22.33_{\pm 0.51}$ | 25.23 |
| MAS [1] | 5 | None | $11.5_{\pm 0.01}$ | $14.45_{\pm 0.01}$ | $9.88_{\pm 0.003}$ | 12.17 | $22.50_{\pm 0.06}$ | $16.89_{\pm 0.07}$ | 19.70 |
| iCaRL [35] | 5 | 1010 | $4.6_{\pm 0.01}$ | $32.01_{\pm 0.14}$ | $23.66_{\pm 0.30}$ | 27.84 | $50.59_{\pm 0.12}$ | $39.68_{\pm 0.003}$ | 45.14 |
| BiC [51] | 5 | 1010 | $1.4_{\pm 0.001}$ | $5.28_{\pm 0.42}$ | $3.39_{\pm 0.09}$ | 4.34 | $6.90_{\pm 0.30}$ | $4.53_{\pm 0.003}$ | 5.72 |
| VilCo | 5 | 1010 | $2.9_{\pm 0.09}$ | $\mathbf{33.58}_{\pm 0.06}$ | $\mathbf{26.24}_{\pm 0.04}$ | **29.91** | $\mathbf{53.75}_{\pm 0.33}$ | $\mathbf{42.70}_{\pm 0.006}$ | **48.23** |

## 3.4 Evaluation Metrics

We evaluate continual learning (CL) models using tailored metrics under a standard CL framework following the suggestions by [16, 30]: **(1) Average performance metrics**: we compare the average performance $P$ to test the ability of the model to adapt to a sequence of video-language tasks. $P$ varies across different tasks, such as MQ, NLQ and VQ. The metric at the $i$-th task is defined as $P_i = \frac{1}{i} \sum_{j=1}^{i} p_{i,j}$ where $P$ represents the cumulative performance up to the current task. $p_{i,j}$ is the performance metric evaluated on the i-th task after the model was trained on the previous $j$ tasks ($j <= i$). In NLQ and MQ tasks, we adopt average recall@k (IoU=m) as the performance metric, where we select top $k = \{1, 5\}$. This metric presents the percentage of query sentences that appear in the top-k predictions with IoU larger than the threshold $m = \{0.3, 0.5\}$. For the VQ task, we leverage temporal AP (tAP) as the performance metric which measures the distance between the predictions and ground-truth localizations. Again we calculate the average of tAP over the previous tasks. **(2) Memory stability metrics**: Following [46], we also consider Backward Forgetting (BwF) to evaluate the performance of CL models. BwF measures the influence caused by learning task $i$ on the performance of the model in remembering previous tasks. $BwF_i$ is calculated as follows: $BwF_i = \frac{1}{i-1} \sum_{j=1}^{i-1} (p_{j,j} - p_{i,j})$. in other words, $p_{i,j}$ represents the performence of the task $j$ after learning the new task $i$. We report total $BwF = BWF_N$ where $N$ is the total number of tasks.

## 4 Evaluation and Discussion

We evaluate the existing continual learning methods on our newly curated benchmark `ViLCo-Bench`We employ several popular continual learning methods, such as EWC [22], MAS [1], iCaRL [35], and BiC [51]. To ensure a fair comparison, all methods utilize the same architectural backbone. For example, we employ EgoVLP-v2 [30] as the visual encoder and CLIP [32] as the text encoder, starting with identical pre-trained parameters for initialization. We report the upper (and lower bound) of the proposed benchmark for each task. These bounds represent multi-task training settings where all samples are available to the model throughout the training process.

### 4.1 Query Incremental challenge

**Moments Queries:** Moments queries focus on identifying and retrieving multiple moments from a long egocentric video in response to a natural language query. Table 2 compares the performance of baseline models and our proposed model within the defined Moment Query (MQ) setup. We used a similar architecture for the upper-bound while keeping the head layers as simple as possible to test the lower-bound performance. The results highlight the impact of replay-based models compared to other baselines (e.g., regularization-based models) because of the complexity of multimodal tasks. Among the continual learning approaches, our proposed method achieved the highest performance in terms of Average Recall. Using self-supervised learning and the episodic memory module, our method improves Avg R@1 (IoU=0.5) by $2.58\%$ with a lower Backward Forgetting (BwF) score.

**Natural Language Queries** This task involves understanding both the semantic meaning of the query and the visual content of the video to accurately localize the relevant moment. Compared with MQ, the NLQ task seems more challenging because the language queries are not limited to human activities but open-vocabulary descriptions. Table 3 provides the comparison. Due to the more complex language descriptions, the average R@1 values decline to more than $30\%$ compared to the previous task. Similarly, Higher BwF values imply that existing CL methods suffer from catastrophic forgetting and performance varies dramatically across tasks. However, using our episodic memory

Table 3: Results of the state-of-the-art and our method on Natural Language query.

| Method | Num. Task | Mem. Capacity | BwF↓ | Avg R@1 (%)↑ | | | Avg R@5 (%)↑ | | |
|---|---|---|---|---|---|---|---|---|---|
| | | | | IoU=0.3 | IoU=0.5 | mean | IoU=0.3 | IoU=0.5 | mean |
| Upper-Bound | None | None | None | 13.82 | 9.20 | 11.51 | 33.59 | 23.18 | 28.39 |
| Naive | 13 | None | 48.76 | 6.05 | 3.61 | 4.83 | 16.77 | 10.07 | 13.42 |
| EWC | 13 | None | 50.05 | 6.34 | 4.05 | 5.20 | 19.50 | 12.08 | 15.79 |
| MAS | 13 | None | 35.92 | 7.04 | 4.22 | 5.63 | 21.56 | 12.63 | 17.10 |
| ViLCo | 13 | 1010 | **10.60** | **9.49** | **6.21** | **7.85** | **25.52** | **16.36** | **20.94** |

Table 4: Results of the state-of-the-art and our method on visual query.

| Method | Num. Task | Mem. Capacity | BwF↓ | Avg tAP$_{25}$ (%)↑ | Avg stAP$_{25}$ (%)↑ | Avg rec (%)↑ | Avg Succ. (%)↑ |
|---|---|---|---|---|---|---|---|
| Upper-Bound | None | None | None | 31 | 22 | 47.05 | 55.89 |
| EWC | 5 | None | 51.01 | 11.48 | 7.81 | 16.79 | 22.05 |
| MAS | 5 | None | 47.60 | 12.13 | 9.16 | 17.80 | 22.51 |
| ViLCo | 5 | 1010 | **23.77** | **17.85** | **13.23** | **26.36** | **33.38** |

Table 5: The impact of various visual features

| Visual Backbone | BwF↓ | Avg R@1 (%)↑ | | Avg R@5 (%)↑ | |
|---|---|---|---|---|---|
| | | IoU=0.3 | IoU=0.5 | IoU=0.3 | IoU=0.5 |
| Timersformer [3] | 2.4 | 30.80 | 22.82 | 51.93 | 40.64 |
| X3D [8] | 1.4 | 31.50 | 23.01 | 48.09 | 36.59 |
| EgoVLP-v2 [30] | **2.9** | **33.58** | **26.24** | **53.75** | **42.30** |

module and SSL mechanism, our method performs better than both replay-based and regularization-based methods, by reducing BwF between 3.5-5 times compared to the other baselines.

**Visual Queries** This task requires a system to understand and interpret both the visual content of a query image and the continuous video stream to locate and return the video segment where the content of the query image appears. As shown in Table 4, it is found that existing methods suffer from catastrophic forgetting and fail to localize accurate objects in later tasks. By proposing episode memory and self-supervised learning, ViLCo facilitates the detection in continuous tasks.

## 4.2 Ablation Study

We investigate the sensitivity to the order of tasks and the ability to remember previous knowledge. We also study the effectiveness of each proposed module (i.e., episodic memory and self-supervised video-language adjustment modules) in the video-language tasks. Due to the space limit, we only present the results on the moments query task, and others are reported in the supplementary material.

**Impacts of various backbones.** Table 5 compares various visual backbones as the vision incoder including EgoVLP-v2, Timersfomer[3] and X3D[8] in extracting effective visual features. Inspired by Mixture of experts (MoE) [39, 19] in multimodal learning, we also evaluated the impacts of different backbones in moment query, such as InternVideo [48], Slowfast [9] and Omnivore [14]. As shown in Table 6, our basic backbone is EgoVLP-v2. We combine EgoVLP-v2 features with the above features by concatenation. It is observed that the combination of EgoVLP-v2 and InternVideo achieves the best performance. Considering that introducing more features increases the difficulty of reducing the gap between video and text representations, more experts are not always better, especially in the continual learning field.

**Effectiveness of proposed modules.** Table 7 compares the influence of proposed modules including the Episodic Memory module (EM) and Self-Supervised Learning (SSL). We provide a base model Naive, removing both the memory module and SSL. Compared with ViLCo w/o SSL, ViLCo improves R@1 (IoU=0.5) by 1.75%, which shows the importance of SSL. Meanwhile, ViLCo using EM only brings marginal gains (comparison between ViLCo w/o EM and ViLCo). However, compared with Naive, ViLCo could further improve the performance through EM and SSL based on the short-term memory buffer. This is because the memory buffer retains a large number of negative samples for contrastive learning and past knowledge for understanding episodes.

**Impact of task orders.** As shown in Table 8, we compare different task-splitting strategies by randomly switching orders of tasks. The results demonstrate that replay-based methods are more sensitive to the order of tasks. Due to imbalanced data distribution in different tasks, replay-based

Table 6: The impact of various visual features. SF(Slowfast [9]) and OV( Omnivore [14])

| Method | Vision Backbone | BwF↓ | Avg R@1 (%)↑ | | | Avg R@5 (%)↑ | | |
|--------|-----------------|------|--------------|---|---|--------------|---|---|
| | | | IoU=0.3 | IoU=0.5 | mean | IoU=0.3 | IoU=0.5 | mean |
| ViLCo | EgoVLP-v2 | 2.9 | 33.58 | 26.24 | 29.91 | 53.75 | 42.70 | 48.23 |
| ViLCo | EgoVLP-v2 + InternVideo | 2.8 | 42.73 | 33.53 | 38.13 | 62.97 | 50.50 | 56.74 |
| ViLCo | EgoVLP-v2 + InternVideo + SF | 4.3 | 38.33 | 29.75 | 34.04 | 56.95 | 46.69 | 51.82 |
| ViLCo | EgoVLP-v2 + InternVideo + SF + OV | 5.59 | 37.79 | 28.18 | 32.99 | 60.94 | 50.24 | 55.59 |

methods are easier to overfit to rare categories or queries if the data occurs in the first tasks. We have provided a more detailed analysis across 6 variations in the order of the tasks in Appendix E

Table 7: Comparing EM (episodic memory) and SSL (self-supervised learning) modules.

| Method | BwF↓ | Avg R@1 (%)↑ | |
|--------|------|--------------|---|
| | | IoU=0.3 | IoU=0.5 |
| Naive | 18.8 | 22.74 | 17.58 |
| ViLCo w/o EM | 4.4 | 32.61 | 25.86 |
| ViLCo w/o SSL | 5.3 | **33.70** | 24.49 |
| ViLCo | **2.9** | 33.58 | **26.24** |

Table 8: The impact of the order of tasks.

| Method | Random Order | BwF↓ | Avg R@1 (%)↑ IoU=0.5 |
|--------|--------------|------|---------------------|
| EWC | 1 | 24.2 | 12.51 |
| iCaRL | 1 | 4.6 | 23.66 |
| ViLCo | 1 | 2.9 | 26.24 |
| EWC | 2 | 15.86 | 18.97 |
| iCaRL | 2 | 6.21 | 20.30 |
| ViLCo | 2 | 1.9 | 23.02 |

## 5 Discussion and Limitations

Our initial experimental analyses have provided several critical insights. First, we observe that previous efforts in continual learning (CL) have primarily focused on uni-modal and class-incremental learning. However, these existing methods suffer from catastrophic forgetting and perform poorly on ViLCo-Bench, particularly regularization-based approaches [22, 1]. The inefficacy arises because each modality has unique learning dynamics, and regularization-based methods apply a uniform constraint across all parameters, which is not effective, leading to diminished performance on prior tasks. Second, although replay-based methods perform better than regularization-based methods, performance degradation also occurs for BiC [51] in the MQ task. BiC aims to correct the bias toward recent tasks, and it may inherently adapt to newer data if the older instances stored in the limited replay buffer are not sufficiently balanced. To resolve possible limited annotation issues, the proposed ViLCo leverages the augmented narration data for SSL. Third, in video-text CL, the data is rich and diverse but the replay buffer could only store a finite amount of past video-text data. The inability to store sufficient examples can lead to incomplete learning and forgetting of critical details essential for the task. Thus, ViLCo develops the EM module to extract task-agnostic cues from previous tasks.

**Limitation.** The current ViLCo architecture leverages fixed backbones and task-specific heads, optimized for predefined CL tasks. However, it lacks a unified framework that could integrate arbitrary tasks in CL such as Visual Question Answering (VQA) [28] and video captioning [44] which require distinct objectives and frameworks. Although our proposed model architecture, ViLCo, does not currently include considerations for those tasks, our curated dataset, ViLCo-Bench, remains highly suitable for a wide range of video-language tasks. Besides architectural limitations, another significant challenge in the current benchmark is the heavy reliance on annotated data streams. Expanding ViLCo-Bench to more modality domains (e.g., audio, images, and time series) exacerbates the requirement for extensive annotations, posing a substantial challenge in real-world applications. Additionally, although data is collected from 9 countries with 74 locations worldwide, half of the videos are from the US, which may introduce societal biases and result in underperformance when applied to other cultural or linguistic groups.

**Future works.** To further extend our benchmark, we would introduce more sophisticated natural language queries and video-language tasks as mentioned above. We will also include more uni-modal data to learn informative representations, such as depth. Moreover, as we focus on egocentric video and language inputs, ViLCo-Bench would be expanded to related applications, such as household robots. However, the influence of introducing different perspectives (e.g., YouTube videos with a third-person perspective) remains relatively unexplored.

**Conclusion.** We proposed `ViLCo-Bench`, a novel benchmark for video-language continual learning. In this work, we defined new tasks and queries based on natural language queries, allowing users to interact with videos through natural language. This approach requires understanding and the textual queries also interpreting the videos and reasoning about the events while considering their temporal relationships. To this end, we propose a memory-efficient benchmark model to investigate the domain gap in video language continual learning. Our curated dataset `ViLCo-Bench` is valuable for multimodal continual learning and would facilitate advancements in the CL field.

## Acknowledgments and Disclosure of Funding

This material is based upon work supported by the International Technology Center Indo-Pacific (ITC-IPAC) under Contract No. FA520923C0020.

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

## A   Related Works on Continual Learning

Continual learning setups described in the literature are mainly divided into three categories: 1) Task-Incremental (Task-IL), 2) Class-incremental (Class-IL), and 3) domain-incremental (Domain-IL). In Task-IL, the model is trained on different tasks with a similar distribution of inputs [18], while in class incremental, the input distribution changes over time [23, 29]. In domain-incremental, on the other hand, the output distribution from one task to the other task remains the same while the input

distribution changes. As the content is provided incrementally in all these scenarios, continual learning is also referred to as incremental learning or lifelong learning in much of the literature, without a strict distinction. This continual/incremental exposure of the model to new data with new distributions leads to the major challenge which is known as **catastrophic forgetting**, where adaptation to the new data generally results in reducing the ability of the model to remember the old patterns. In fact continual learning is a trade-off between two challenges: "memory stability" and "learning plasticity". One naive solution to this is to repeat the training process for all the current and previous data, however, this is not possible due to limitations in resources, or missing access to previous data because of privacy. This emphasizes the need for continual and online learning methods. Existing CL approaches to these challenges can be divided into: • Regularization-based approach: adding regularization terms with reference to the old model to penalize abrupt changes in the most relevant parameters learned from previous tasks such as Elastic Weight consolidation (EWC) [22]. • Replay(Rehearsal)-based: rehearsing and recovering the previous distributions [38, 46]. • Architecture-based: customizing architecture to accommodate new tasks by freezing some parameters or sub-layer/networks of the model (WSN[21] or extending the architecture(iCaRL[35], BiC[51]). • Distilliation-based: Retaining already inference knowledge from previous rounds of training by storing the models or logits and reusing them during new training rounds [5, 23]. We refer to [47] provides a comprehensive study and review of existing continual learning approaches.

**Continual Learning (CL) in Video** In addition to regular challenges of the CL problems, Continuous learning from videos comes with its own unique set of challenges such as: 1) Considering the temporal dependency within video clips or video streams. 2) Necessity of detecting temporal boundary as well as spatial boundary as the activity of interest can happen in any subsegment of the given video. 3) Increase the amount of Noise and irrelevant data, 4) Given the resolution and length of the video, most approaches are not scalable to larger than 2 seconds video. Additionally, the performance of replay-based methods degenerates substantially due to the limited memory space.

PIVOT [45] exploits an image-language pre-trained model (CLIP [32]) and extends it temporally to consider video clips. Simultaneously, [2] improved vCLIMB [46] rehearsal strategy by storing only a single form of videos for replying purposes. Both methods showed they outperform conventional video CIL methods across the same test bed. Efficient-CLS [50] is another state-of-the-art video online continual learning (OCL) model proposed for low-labeled scenarios such as learning from online streaming videos.

However, there are a few challenges that are not addressed by any of the proposed methods so far. First, they only focus on class-incremental tasks with the final task of classification. Second, None of them consider multimodal continual learning tasks such as moment query, natural language query and vision query tasks. However, existing works use different evaluation protocols, making direct comparisons between methods difficult. Moreover, these works mostly propose to pre-train the model with a large set of classes of the same data distribution (up to half of the total) which is against real-world situations [46]. Additionally, all the existing models, consider a special case of CIL assuming each sample video clip corresponds to a single label, while in real-world scenarios video clips are usually related to multiple labels or even multiple actions during their time. Hence MultiLabel Class-Incremental Learning is an important task in video CL approaches.

## B   NLQ Query Templates

1. Objects: What did I put in X?
2. Place: Where did I put X?
3. Objects: Where is object X before/after event Y?
4. People: When did I talk to or interact with person with role X?
5. Objects: How many X's? (quantity question)
6. Objects: State of an object
7. Objects: Where is object X
8. Objects: In what location did I see object X ?
9. Objects: What X did I Y?
10. Objects: What X is Y?

11. Objects: Where is my object X?

12. People: Who did I interact with when I did activity X?

13. People: Who did I talk to in location X?

## C  SSL with Narations

We denote the objective loss as:

$$\mathcal{L}_{v2t} = -\frac{1}{B}\sum_i^B \log \frac{exp(sim(V_i, T_i))}{\sum_{j=1}^B exp(sim(V_i, T_j))}, \tag{1}$$

$$\mathcal{L}_{t2v} = -\frac{1}{B}\sum_i^B \log \frac{exp(sim(V_i, T_i))}{\sum_{j=1}^B exp(sim(V_j, T_i))}, \tag{2}$$

$$\mathcal{L} = \mathcal{L}_{v2t} + \mathcal{L}_{t2v}, \tag{3}$$

where $V_i$ is the video embedding, while $T_i$ is the textual representations of narrations. We consider $V_i$ and its corresponding $T_i$ as the positive pairs.

## D  Training Details

Table 9: Task-specific training details.

| Task | Moments Query | Natural Language Query | Visual Query |
|------|---------------|------------------------|--------------|
| Task Details | | | |
| Query Type | action categories | natural language | image of an object |
| Num. of Tasks | 5 | 13 | 5 |
| Num. of Train Queries | 15511 | 13849 | 13692 |
| Num. of Val Queries | 4932 | 4554 | 4527 |
| Hyperparameters | | | |
| Initial Learning Rate | 0.0001 | 0.0001 | 0.0003 |
| Weight Decay | 0.05 | 0.05 | 0.0001 |
| Num. Epochs per Task | 15 | 13 | 50 |
| Batch Size | 2 | 8 | 4 |

## E  Further Analysis on the Impact of the Order of Tasks.

From Figure 5, we observe varying performance based on the order of exposure to different data/sub-tasks. It is important to note that each Task $i$ (ranging from 1 to 5) incorporates different action categories and sample sizes. Sub-tasks with limited data tend to exhibit long-tail distributions, which can cause models to overfit if such tasks are introduced early, leading to performance degradation in subsequent tasks. This phenomenon explains why ViLCo exhibits a higher Avg. R@1 with increased BWF, highlighting that task ordering is a significant source of variance in continual learning.

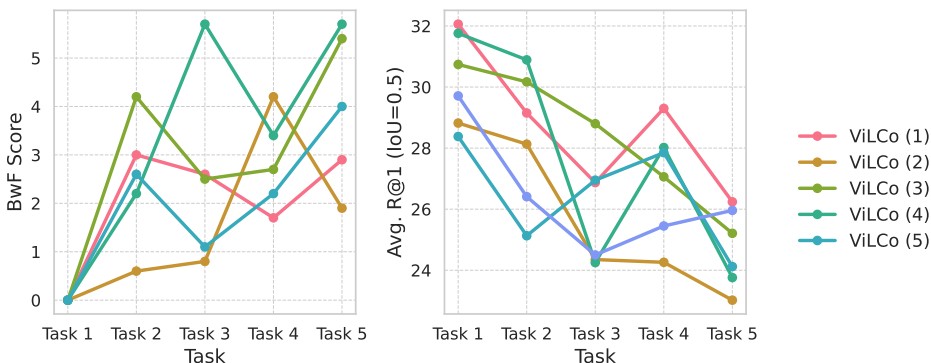

Figure 5: Impact of the order of tasks in Moment Query scenario: BwF (left) and Average Recall (Avg. R@1) (right)

