# `ViLCo-Bench`: VIdeo Language COntinual learning Benchmark

**Tianqi Tang**[1], **Shohreh Deldari**[1], **Hao Xue**[1], **Celso De Melo**[2], **Flora Salim**[1]

1 School of Computer Science Engineering, University of New South Wales, Australia
2 DEVCOM Army Research Laboratory, US
[1] {tianqi.tang, s.deldari, hao.xue1, flora.salim}@unsw.edu.au
[2] celso.miguel.de.melo@gmail.com

## 1  Datasheet

### 1.1  Motivation

1. **For what purpose was the dataset created?** Multimodal continual learning is vital for efficient AI agents. The existing continual learning community lacks a comprehensive benchmark specifically designed for multimodal applications and to evaluate across various multimodal tasks. To address this, we propose ViLCo-Bench.

2. **Who created the dataset(e.g., which team, research group) and on behalf of which entity (e.g., company, institution, organization)?** The dataset is created by Tianqi Tang, Shohreh Deldari, Hao Xue, and Flora Salim from the University of New South Wales, Australia, and Celso De Melo from DEVCOM Army Research Laboratory, US.

3. **Who funded the creation of the dataset?** The project is funded by the Army Research Laboratory US.

### 1.2  Composition

1. **What do the instances that comprise the dataset represent (e.g., documents, photos, people, countries)?** The instances are video clips along with their notation extracted from the Ego4D dataset (`https://ego4d-data.org/`). The extracted video clips are employed to generate samples suitable for training video-language-specific tasks.

2. **How many instances are there in total (of each type, if appropriate)?** See Table below:

Table 1: Task-specific samples.

| Task | Moments Query | Natural Language Query | Visual Query |
|---|---|---|---|
| Num. of Tasks | 5 | 13 | 5 |
| Num. of Samples | 20443 | 18403 | 18219 |

3. **Does the dataset contain all possible instances or is it a sample (not necessarily random) of instances from a larger set?** Our dataset is a novel dataset for video-language continual learning, which includes all specific metadata, video clips and annotations.

4. **What data does each instance consist of?** Each instance consists of the video clip, corresponding label(s) and narrations, and query (either in text or image).

5. **Is there a label or target associated with each instance?** The labels can vary depending on the type of query. For Moment Queries and Natural Language Queries, the labels are in

Submitted to the 38th Conference on Neural Information Processing Systems (NeurIPS 2024) Track on Datasets and Benchmarks. Do not distribute.

the form of one or multiple ranges of timeslots within the corresponding video. Additionally, for Vision Queries, the labels are bounding boxes specifying the queried object in the input video clip.

6. **Is any information missing from individual instances? If so, please provide a description, explaining why this information is missing (e.g., because it was unavailable). This does not include intentionally removed information but might include, e.g., redacted text.** Not applicable.

7. **Are relationships between individual instances made explicit (e.g., users' movie ratings, social network links)?** Not Applicable

8. **Are there recommended data splits (e.g., training, development/validation, testing)?** There are no specific requirements here, however, we have published the data splits based on tasks, training and validation sets. We also provided the corresponding data loaders in our benchmark's GitHub Repository.

9. **Is the dataset self-contained, or does it link to or otherwise rely on external resources (e.g., websites, tweets, other datasets)?** It is self-contained. However, to extract features, we have used other existing resources which are mentioned in the GitHub repository.

10. **Does the dataset contain data that might be considered confidential (e.g., data that is protected by legal privilege or by doctor-patient confidentiality, data that includes the content of individuals' non-public communications)?** No.

11. **Does the dataset contain data that, if viewed directly, might be offensive, insulting, threatening, or might otherwise cause anxiety?** No.

12. **Does the dataset identify any subpopulations (e.g., by age, gender)?** No.

13. **Is it possible to identify individuals (i.e., one or more natural persons), either directly or indirectly (i.e., in combination with other data) from the dataset?** In some of the video clips, the face of people around the camera are visible. However, since we have not collected the data and we have only used already published videos in `https://ego4d-data.org/`.

14. **Does the dataset contain data that might be considered sensitive in any way (e.g., data that reveals race or ethnic origins, sexual orientations, religious beliefs, political opinions or union memberships, or locations; financial or health data; biometric or genetic data; forms of government identification, such as social security numbers; criminal history)?** No.

## 1.3 Collection Process

1. **How was the data associated with each instance acquired?** Not Applicable

2. **What mechanisms or procedures were used to collect the data (e.g., hardware apparatuses or sensors, manual human curation, software programs, software APIs)?** We utilize Ego4d APIs to gather original video clips and segments of visual features. Furthermore, we employ Python scripts to extract features and configure settings for continual learning.

3. **If the dataset is a sample from a larger set, what was the sampling strategy (e.g., deterministic, probabilistic with specific sampling probabilities)?** The original video clips are sourced from the public Ego4d dataset. We employ a deterministic sampling strategy and make use of all available videos along with their annotations from NLQ, MQ, and VQ.

4. **Who was involved in the data collection process (e.g., students, crowdworkers, contractors) and how were they compensated (e.g., how much were crowdworkers paid)?**
Not Applicable

5. **Over what timeframe was the data collected?**
Not Applicable

6. **Were any ethical review processes conducted (e.g., by an institutional review board)?** Not Applicable

7. **Did you collect the data from the individuals in question directly, or obtain it via third parties or other sources (e.g., websites)?** The data is obtained from the original publisher with their permission.

8. **Were the individuals in question notified about the data collection?** Yes.

9. **Did the individuals in question consent to the collection and use of their data?** Yes.

10. **If consent was obtained, were the consenting individuals provided with a mechanism to revoke their consent in the future or for certain uses?** The consent and the revoking procedure is handled by the Ego4D dataset publishers. However, we adhere to apply any required changes.

11. **Has an analysis of the potential impact of the dataset and its use on data subjects (e.g., a data protection impact analysis) been conducted?** Not Applicable

## 1.4 Preprocessing/cleaning/labeling

1. **Was any preprocessing/cleaning/labeling of the data done (e.g., discretization or bucketing, tokenization, part-of-speech tagging, SIFT feature extraction, removal of instances, processing of missing values)?** Yes, our dataset employs a distinct data pre-processing and splitting approach. For more information, please see Section 3.2. Additionally, we have excluded a small number of videos that lack annotations from our benchmark.

2. **Was the "raw" data saved in addition to the preprocessed/cleaned/labeled data (e.g., to support unanticipated future uses)?** Yes, we provide the download link of "raw" data on our website, including pre-extracted features and original video clips.

3. **Is the software that was used to preprocess/clean/label the data available?** Yes. We made all the code publicly accessible in our benchmark GitHub repository.

## 1.5 Uses

1. **Has the dataset been used for any tasks already?** No, the dataset is newly proposed by our team.

2. **Is there a repository that links to any or all papers or systems that use the dataset?** Yes, please refer to the link (`https://github.com/cruiseresearchgroup/ViLCo`).

3. **What(other) tasks could the dataset be used for?** Following newly proposed query-incremental learning, we can expand our dataset for other video-language tasks within continual learning, such as Visual Question Answering (VQA) and moment retrieval.

4. **Is there anything about the composition of the dataset or the way it was collected and preprocessed/cleaned/labeled that might impact future uses?** We do not believe so because the ViLCo dataset is collected from a high-quality public egocentric Ego4d dataset. We plan to extend our dataset with the new version, and we will also update the website and the corresponding document for future uses.

5. **Are there tasks for which the dataset should not be used?** No.

## 1.6 Distribution

1. **Will the dataset be distributed to third parties outside of the entity (e.g., company, institution, organization) on behalf of which the dataset was created?** No.

2. **How will the dataset will be distributed (e.g., tarball on website, API, GitHub)?** The dataset will be publicly accessed on our website.

3. **When will the dataset be distributed?** We release the dataset at the Zenodo under DOI: 10.5281/zenodo.11560095. We also release the corresponding code at `https://github.com/cruiseresearchgroup/ViLCo`.

4. **Will the dataset be distributed under a copyright or other intellectual property (IP) license, and/or under applicable terms of use (ToU)?** We release our dataset under Creative Commons Attribution 4.0 International License.

5. **Have any third parties imposed IP-based or other restrictions on the data associated with the instances?** No.

6. **Do any export controls or other regulatory restrictions apply to the dataset or to individual instances?** No.

### 1.7 Maintenance

1. **Who will be supporting/hosting/maintaining the dataset?** The University of New South Wales (UNSW) Collective + Robust Ubiquitous Sensing + Intelligence (CRUISE) team.

2. **How can the owner/curator/manager of the dataset be contacted (e.g., email address)?** We provide email addresses for the corresponding authors.

3. **Is there an erratum?** No. If errors are found in the future, we will provide an erratum in the webpage of our benchmark (`https://github.com/cruiseresearchgroup/ViLCo`).

4. **Will the dataset be updated (e.g., to correct labeling errors, add new instances, delete instances)?** Yes. If there exist labeling errors, we will update the dataset with the new version. We also have a plan for extending the benchmark with more modalities, queries and tasks.

5. **If the dataset relates to people, are there applicable limits on the retention of the data associated with the instances (e.g., were the individuals in question told that their data would be retained for a fixed period of time and then deleted)?** No.

6. **Will older versions of the dataset continue to be supported/hosted/maintained?** Yes, the older versions of the dataset continue to be supported/hosted/maintained.

7. **If others want to extend/augment/build on/contribute to the dataset, is there a mechanism for them to do so?** We invite researchers to enhance our dataset by adding new tasks or annotations. For the consistency and reliability of the additional data, we recommend adhering to our sampling strategy and metadata framework, detailed at `https://github.com/cruiseresearchgroup/ViLCo`. Contributors can submit their annotations via a pull request. Upon verification of these new annotations, we will integrate them into our dataset and duly acknowledge the contributions of the researchers involved.

### 1.8 Computing Resource

We used multiple NVIDIA V100 nodes in a high-performance computing cluster, called National Computational Infrastructure (NCI) in Australia.