# OpenReview forum: "ViLCo-Bench: VIdeo Language COntinual learning Benchmark"
_NeurIPS.cc/2024/Datasets_and_Benchmarks_Track — NeurIPS 2024 Track Datasets and Benchmarks Poster_

### Official Review · Reviewer_NAFV · 2024-07-18
**Paper is hard to parse due to lack of details and citations in several spots**

**Rating:** 7
**Confidence:** 4

**Review:**

The authors seem to have constructed a solid benchmark, given clear guidelines and a concrete blueprint, leveraging the language capabilities of the tested models to move away from simply introducing new classes to more nuanced progressive changes. However, there are multiple instances in the manuscript where there are vital citations missing, or important details lacking, lessening the potential impact of the paper. The work seems to be original in conception, though because only a single dataset is included in the continual learning benchmark, this originality, but also the significance of it come into question.

**Strengths:**

The paper tries to outline how the dataset is constructed, and the methods the authors used to balance the different segments. The authors also introduce a framework that seems to outperform previous work on their benchmark, an important contribution onto itself. This work serves as an important reminder of contemporary challenges in deep learning and foundation models, and perhaps a new frontier in video-language tasks.

**Additional Feedback:**

I am willing to slightly raise my rating provided my concerns for the authors’ repo are addressed, so I ask them not to be discouraged by the current borderline rejection score. Nevertheless, my updated rating will also take into account other reviews and the ensuing discussion.

**Clarity:**

The papers unfortunately has several sections with missing citations or complex commentary. As examples, I mention:
- lines 55-57: citations needed
- lines 118-121: citations and better explanation of their assertions needed
- lines 144-148: I am not sure how repeating a bicycle is relevant to a crowded airport
- lines 153-155: assertions need better explanation and citations of previous work showing evidence for them
- lines 166-173: this paragraph is really hard to parse, and it talks about an very important topic. Consider clarifying or adding a section in the appendix.
- lines 240-248: better explanation and motivation needed (the formulas take time to understand, and therefore clearer commentary would make them much easier to parse)
- Figure 4 needs a lot of work; as of now, text overlaps with shape boundaries

**Correctness:**

The dataset seems to have been constructed in a sound way, making it challenging for video-language models.

**Documentation:**

The paper comes with a repo, yet more details are needed in the README for the handling and the construction of the dataset, as well as a precise link to the dataset, and not a generic link to the front page of the website hosting it. This is fundamental to the success and the usefulness paper, and is one of the main reasons I am advocating for borderline rejection at this time.

**Limitations:**

The authors do not really mention the above challenges in their limitations (a brief mention can be said to be made in line 339, yet that captures only a fraction of the potential distribution shifts, and does not address perhaps more pertinent problems in domains closer to Ego4D, but rather directly jumps to third-person, which admittedly is also relevant), yet they cite previous work in vision-language benchmarks that has tackled these challenges. The authors also do not seem to pay enough attention to mixing the 3 different tasks, which is also an important aspect of a useful continual learning agent.

**Opportunities For Improvement:**

The benchmark contains instances from a single dataset. That means that the data distribution remains stagnant between different segments, potentially compromising the promise of the paper to evaluate continual learning. For example, Climb [36] integrated several different benchmarks into their framework, allowing researchers to show not just how generalizable a model is to new tasks, but also new data distributions. That is to say, the encoders might not have to deal with a significant distribution shift in the input, which does not capture the entire gamut of potential challenges of continual learning and real-world deployment of machine learning systems.

**Relation To Prior Work:**

The authors seem to providing enough background discussion, though a citation seems to be off in line 31 (should be [36] to the the best of my understanding, not [40], as [40] contains video). There also seem to be some bugs in some citations in the appendix.

**Summary And Contributions:**

The authors introduce a continual learning video-language benchmark based on Ego4D. The benchmark is designed to be very challenging, moving away from just introducing more classes to incorporating different kinds of questions, etc. Moreover, the tasks contained within are inherently challenging. The benchmark therefore can serve even as a challenging common-sense video-language dataset for future video-language models.

Note: rating has been revised to take into account the rebuttal and the updates provided by the authors.

---

> ### Author Rebuttal · Authors · 2024-08-16
>
> We sincerely thank the reviewer for taking the time to provide valuable suggestions and detailed comments. We are pleased that the reviewer found our benchmark to be solid and concrete. We have taken your comments seriously. We believe that your thorough comments have significantly enhanced the quality of our work.  Looking forward to further discussions with you.
>
> ### Limitations:
>
> **Q1 & L1: Discussion on Limited Data Distribution Shift**
> > The authors do not really mention the above challenges in their limitations (a brief mention can be said to be made in line 339, yet that captures only a fraction of the potential distribution shifts, and does not address perhaps more pertinent problems in domains closer to Ego4D, but rather directly jumps to third-person, which admittedly is also relevant), yet they cite previous work in vision-language benchmarks that has tackled these challenges.
>
> The MQ, NLQ, and VQ tasks involve distinct video clips with different objectives. Additionally, the original Ego4D dataset is the largest of its kind, covering 74 locations globally. To enhance data diversity in our benchmark, we've provided guidelines for adding new data/tasks, available [here](https://github.com/cruiseresearchgroup/ViLCo/blob/main/Add_New_Task.md). Following these guidelines, we integrated new data from MSRVTT-QA for VQA tasks. The evaluation results are shown below.
>
> **Table 4. Performance Comparison with Different CL Methods on MSRVTT-QA**
>
> | Method | BwF | Accuracy |
> |--------|-----|----------|
> | EWC    | 8.53| 17.3%    |
> | iCaRL  | 3.8 | 21.6%    |
> | ViLCo  | 2.4 | 24.8%    |
>
> **L2: Evaluating the Mixture of Three Different Tasks (MQ, NLQ, and VQ)**
> > The authors also do not seem to pay enough attention to mixing the 3 different tasks, which is also an important aspect of a useful continual learning agent.
>
> Unlike typical class-incremental learning, combining vastly different tasks with varying modalities, output formats, and objectives introduces challenges that limit the effectiveness of evaluating continual learning in our framework. Below are the key reasons:
>
> - **Task Heterogeneity:** Handling multiple distinct input modalities and output types complicates the learning process and makes it difficult for a model to manage these differences effectively.
> - **Catastrophic Forgetting:** Switching between unrelated tasks, like video localization and text retrieval, increases the risk of catastrophic forgetting as the model needs to continually adjust to very different objectives with distinct headers.
> - **Lack of Transferability:** CLiMB [36] adopts a few-shot continual learning setting. In our case, MQ, NLQ, and VQ have limited commonalities, leading to minimal knowledge transfer between them. This reduces the benefits of query-incremental continual learning since the model may need to essentially learn each task from scratch.
> - **Evaluation Difficulties:** Evaluating such diverse tasks together is challenging since standard benchmarks and metrics are designed for more homogeneous tasks, leading to potentially misleading or inconsistent results.
>
> ### Clarity:
>
> We have thoroughly revised the manuscript, making extensive modifications to the writing. All points raised by the reviewer have been carefully addressed. Additionally, we updated Figure 4 to clarify the architecture.
>
> Point-by-point response:
> > lines 55-57: citations needed
>
> We added references [1, 2, 3] after "the uni-modal domain" and reference [4] after "video-language CL tasks".
>
> > lines 118-121: citations and better explanation of their assertions needed
>
> Changing to 'Episodic memory refers to the ability to remember specific experiences and make past videos queriable, such as "what did I eat" and "who did I sit by on my first flight to France?", which are different from semantic memory tasks like "What’s the capital of France?"'
>
> > lines 153-155: assertions need better explanation and citations of previous work showing evidence for them
>
> Changing to 'These multimodal tasks introduce new challenges compared to unimodal (vision-only) models such as noisy videos due to quick head movements and limited field of view, free-form queries in natural language, and tiny response windows within lengthy video footage [5]'
>
> > lines 166-173: this paragraph is really hard to parse, and it talks about an very important topic.
>
> Changing to 'To avoid the label overlapping issue across tasks and samples, we adopted a partitioning strategy to create and split training sets for different tasks and sub-tasks. Since over 90% of videos contain multiple classes spanning different subsets, we ensured each video was assigned to only one subset. We did this by prioritizing higher-frequency queries and excluding those that appeared in multiple subsets, thereby minimizing overlap and enhancing the clarity of learning objectives. This approach supports more effective training and evaluation of models in continual learning scenarios.'
>
> > lines 240-248: better explanation and motivation needed
>
> Changing to 'This metric presents the percentage of query sentences that appear in the top-k predictions with IoU larger than the threshold $m=\{0.3, 0.5\}$. For the VQ task, we leverage temporal AP (tAP) as the performance metric which measures the distance between the predictions and ground-truth localizations. Again we calculate the average of tAP over the previous tasks. **(2) Memory stability metrics**: Following [1], we also consider Backward Forgetting (BwF) to evaluate the performance of CL models. BwF measures the influence caused by learning task $i$ on the performance of the model in remembering previous tasks. $BwF_{i}$ is calculated as follows: $BwF_{i}=\frac{1}{i-1} \sum_{j=1}^{i-1} (p_{j,j} - p_{i,j})$. in other words, $p_{i,j}$ represents the perforemance of the task $j$ after learning the new task $i$. We report total $BwF=BWF_{N}$ where $N$ is the total number of tasks.'
>
> > Figure 4 needs a lot of work
>
> The revised figure is attached (PDF).

---

> > ### Author Response · Authors · 2024-08-21
> > **Official Comment by Authors**
> >
> > Dear reviewer NAFV, we sincerely appreciate the time and effort you invested in providing such valuable feedback on our work. We have carefully addressed all your comments and made substantial revisions based on your suggestions. We kindly ask that you review our updates, as we are eager to hear your thoughts. Detailed responses and additional experiments can be found in the above rebuttal for reviewer NAFV and the global rebuttal at the top of the page. We truly look forward to further discussions and to continuously improving the manuscript. Thank you once again for your support and insightful feedback.

---

> > > ### Comment · Reviewer_NAFV · 2024-08-26
> > > **Comments after revision**
> > >
> > > Dear authors, I appreciate the way you have addressed my call to improve the repo of the benchmark. It is now full of details, and the link seems to be working fine. I am still not convinced about the mixing of tasks: the difficulties you cite seem to be well-suited for a challenging benchmark to push the field forward. Nonetheless, it is possible for researchers to mix the tasks in the future, and therefore is not a major blocking point. I will revise my rating appropriately. I just want to note that lines 241-242 are still a bit ambiguous as to its cumulative nature. Has the network been trained for all tasks up to i, and you are checking how well the tasks are remembered? That seems to be the logical conclusion, but even more clarity to make the segment and the formulas understandable on the first read would be appreciated. Thank you for the thoroughness of your responses.

---

> > > > ### Author Response · Authors · 2024-08-27
> > > > **Clarification on Evaluation Formulas**
> > > >
> > > > Dear Reviewer,
> > > >
> > > > We would like to thank you for taking the time to read our comments and revisions. We are delighted you have seen value in our work and increased your rating.
> > > > Sorry, our previous answer did not completely show our revisions of Section 3.4.  Please find our revised paragraph below. We hope this clarifies the formula.
> > > >
> > > > >  I just want to note that lines 241-242 are still a bit ambiguous as to its cumulative nature. Has the network been trained for all tasks up to i, and you are checking how well the tasks are remembered? That seems to be the logical conclusion, but even more clarity to make the segment and the formulas understandable on the first read would be appreciated. Thank you for the thoroughness of your responses.
> > > >
> > > > Here is the our revised paragraph:
> > > >
> > > > We evaluate continual learning (CL) models using tailored metrics under a standard CL framework following the suggestions by [14,28]:
> > > >
> > > >  **(1) Average performance metrics:** We compare the average performance $P$ to test the ability of the model to adapt to a sequence of video-language tasks. $P$ varies across different tasks, such as MQ, NLQ and VQ. The metric at the $i$-th task is defined as $P_{i}=\frac{1}{i} \sum_{j=1}^{i} p_{i,j}$
> > > > where $P$ represents the cumulative performance up to the current task.
> > > > $p_{i,j}$ is the performance metric evaluated on the i-th task after the model was trained on the previous $j$ tasks ($j <= i$).
> > > > In NLQ and MQ tasks, we adopt average recall@k (IoU=m) as the performance metric, where we select top $k=\{1, 5\}$.
> > > > This metric presents the percentage of query sentences that appear in the top-k predictions with IoU larger than the threshold $m=\{0.3, 0.5\}$. For the VQ task, we leverage temporal AP (tAP) as the performance metric which measures the distance between the predictions and ground-truth localizations. Again we calculate the average of tAP over the previous tasks.
> > > >
> > > > **(2) Memory stability metrics:** Following [44], we also consider Backward Forgetting (BwF) to evaluate the performance of CL models. BwF measures the influence caused by learning task $i$ on the performance of the model in remembering previous tasks. $BwF_{i}$ is calculated as follows: $BwF_{i}=\frac{1}{i-1} \sum_{j=1}^{i-1} (p_{j,j} - p_{i,j})$. in other words, $p_{i,j}$ represents the perforemance of the task $j$ after learning the new task $i$. We report total $BwF=BWF_{N}$ where $N$ is the total number of tasks.

---

### Official Review · Reviewer_ifvZ · 2024-07-24
**ViLCo-Bench review**

**Rating:** 6
**Confidence:** 3
**Correctness:** Seems correct.
**Clarity:** Clear enough.

**Review:**

I think this dataset will be very useful for learning VL CL models. The dataset clearly distinguishes itself from previous ones with single modality or image-focus. The size of the dataset is not very large, but it is considered sufficient for many downstream applications.

**Strengths:**

- VL CL dataset for multiple tasks mainly for episodic memory
- multimodal focus

**Additional Feedback:**

No additional feedback.

**Documentation:**

Yes, sufficient details.

**Ethics:**

No issues detected.

**Limitations:**

Size of the dataset might be one concern.

**Opportunities For Improvement:**

Writing can be improved as it seems overly complicated at places.

**Relation To Prior Work:**

Good.

**Summary And Contributions:**

This paper introduced ViLCo-Bench, a dataset for continual learning for video-text tasks. It contains videos of ~10 mins long and language queries from Ego4d.  The benchmark contains three CL tasks for long vidieos: moments queries, natural language queries, and visual queries.

---

> ### Author Rebuttal · Authors · 2024-08-16
>
> We thank the reviewer for taking the time to assess our manuscript and offering valuable suggestions. We are excited to hear that the reviewer found our benchmark useful for the community. Following the reviewer’s feedback, we have updated our manuscript accordingly. Please let us know if any of our responses are unclear or if a new question or comment comes up.
> Here are point-by-point responses:
>
>
> ### Limitation: Size of the dataset
>
> > Size of the dataset might be one concern.
>
> We appreciate the reviewer's acknowledgment of the dataset’s impact, as noted in your comment: “The size of the dataset is not very large, but it is considered sufficient for many downstream applications.”
>
> We would like to draw attention to Figure 2 in the manuscript, where we compare our proposed benchmark with other existing continual learning benchmarks in terms of dataset size and the number of tasks. This figure highlights the significant contribution of ViLCo-Bench to the field. While we recognise that the dataset size may be smaller compared to some current vision datasets, it is still substantially larger than existing benchmarks in the area of **continual learning**, as you have also observed.
>
> We are considering including more data to extend the impact of the work by introducing new kinds of tasks, categories, and data distributions. The instructions on how to add data are as follows:
> + https://github.com/cruiseresearchgroup/ViLCo/blob/main/Add_New_Task.md
>
> ### Opportunities for Improvements
> > Writing can be improved as it seems overly complicated at places.
>
> Thank you for your constructive feedback. We have carefully proofread the manuscript and revised the ambiguous parts while addressing all the points raised by reviewer NAFV18. We specially:
>   * Updated Section 3: including Methodology section, Dataset curation, model architecture, and evaluation metrics.,
>   * Incorporated additional citations to ensure a clearer understanding of the proposed method and position our work among other SOTAs.
>   * Enhanced Figure 4 by clarifying the architecture details and eliminating any overlaps between text and shape boundaries.  The updated figure is attached.
>   * Other edits and proofreadings across the manuscript.

---

> > ### Author Response · Authors · 2024-08-21
> > **Official Comment by Authors**
> >
> > Dear reviewer ifvZ, thank you once again for your valuable feedback. We hope our rebuttal has addressed your comments. If there are any further clarifications or adjustments you would suggest, please do let us know. We would be grateful for any additional input to help us refine our work as much as possible before the rebuttal period ends.

---

### Official Review · Reviewer_EZFs · 2024-07-26
**CL Benchmark for videos introducing Query Incremental Learning.**

**Rating:** 8
**Confidence:** 3
**Correctness:** The benchmark and evaluation seem to …
**Clarity:** Overall well written

**Review:**

The focus on video-text CL tasks is an interesting approach.
The authors mark their main contribution, the standardized benchmark in multimodal continual learning for video data while defining protocols for training and metrics for proper evaluation.
The five parts of the ViLCo Model were introduced as: Video Feature Encoder, Textual Encoder, Cross-modal Encoder, Task-specific Head, and Episodic Memory Module. Also, Self-supervised learning with narrations was used to improve the cross-modal representations.
The authors also discuss the classic situation of catastrophic forgetting, and Table 4 reports that the use of episode memory and self-supervised learning improves/facilitates detection.

**Strengths:**

ViLCo-Bench proposes a benchmark for Video-Text Continual Learning (CL), which could provide a standardized framework for evaluating model performance in visual-text multimodal CL settings. This benchmark may facilitate more consistent comparisons between different methodologies in the field. Such a resource could potentially contribute to the advancement of research in this area by offering a common evaluation platform.

The paper provides empirical results suggesting potential benefits from integrating visual and textual modalities. The authors' findings indicate that this multimodal approach may lead to more robust representations, potentially mitigating catastrophic forgetting in continual learning scenarios. However, further investigation may be necessary to fully substantiate these claims and understand the extent of these benefits across various conditions.

**Additional Feedback:**

N/A

**Documentation:**

Code and project available on Github:
Dataset available on Zenodo: https://zenodo.org/records/11560095 [I was able to download it but haven't opened it since it requires download all the piece to able to unzip it.]

**Limitations:**

The current limitations for tasks like VQA and video captioning are not supported by the ViLCo method.
Future works address more diversity and the possibility of more sophisticated natural language; what is still unclear is how the current limitation could be investigated or addressed in the future.

**Opportunities For Improvement:**

Can the authors explain a bit better about the random order in which the tasks work better and why this improves your results?
Extend your ablation studies. Especially in CL. Have the authors considered any other backbones for the visual features, such as TimeSformer, X3D, ViViT, MViT, or VideoMAE?

**Relation To Prior Work:**

Table 1 points out the most related benchmarks such as CLiMB although vCLIMB is the closest to the video data type.

**Summary And Contributions:**

The authors introduced ViLCo-Bench, a benchmark for video language continual learning.

Main contributions can be summarized as:

- Developing a standardized benchmark for multimodal continual learning with video data, featuring protocols for training and evaluation metrics.
- Three multimodal tasks in a continual learning setup: Moment Query (MQ), Natural Language Query (NLQ), and Visual Query (VQ).
- A curated dataset based on Ego4D, for multimodal continual learning tasks.
- The authors propose a memory-efficient framework that incorporates self-supervised learning to address challenges such as memory complexity from long video clips, natural language complexity, and text-video misalignment.
- Provide comparative analysis against four state-of-the-art models in video and continual video learning for each benchmark setup.

---

> ### Author Rebuttal · Authors · 2024-08-16
>
> We sincerely thank the reviewer EZFs26 for the constructive comments and suggestions. We are encouraged that the reviewer acknowledges that video-text continual learning problem is well-motivated and new insights in the benchmark can contribute to the relative field.
>
> **Q1: Analysis of the effects of random ordering.**
> > Can the authors explain a bit better about the random order in which the tasks work better and why this improves your results?
>
> We explore this aspect in the last part of Section 4.2, specifically analyzing different task orders and their influence on performance. Table 7 presents the final performance (i.e., in terms of BwF and IoU) of the model taking different path of training by being exposed to different orders of sub-tasks for MQ scenario.
>
> To further validate the impact of the order of tasks and clarify the results, we present additional experiments in the tables below, comparing both the BWF (Backward Forgetting) and Average Recall (Avg. R@1) with an Intersection over Union (IoU) threshold of 0.5. Each run indicates a new random task ordering.
>
> **Table 1. BwF Comparison in Different Sub-Tasks on MQ.**
> | Model | Task 1 | Task 2 | Task 3 | Task 4 | Task 5 |
> |--------|--------|--------|--------|--------|---------------|
> | ViLCo (run 1) | N/A    | 3.0    | 2.6    | 1.7    | 2.9    |
> | ViLCo (run 2) | N/A    | 0.6    | 0.8    | 4.2    | 1.9    |
> | ViLCo (run 3) | N/A    | 4.2    | 2.5    | 2.7    | 5.4    |
> | ViLCo (run 4) | N/A    | 2.2    | 5.7    | 3.4    | 5.7    |
> | ViLCo (run 5) | N/A    | 2.6    | 1.1    | 2.2    | 4.0    |
> | ViLCo (run 6) | N/A    | 3.3    | 5.2    | 4.2    | 2.5    |
>
> **Table 2. Avg. R@1 (IoU=0.5) Comparison in Different Sub-Tasks on MQ**
> | Model | Task 1 | Task 2 | Task 3 | Task 4 | Task 5 |
> |--------|--------|--------|--------|--------|------------|
> | ViLCo (run 1) | 32.06  | 29.15  | 26.87  | 29.30  | 26.24  |
> | ViLCo (run 2) | 28.82  | 28.13  | 24.35  | 24.26  | 23.02  |
> | ViLCo (run 3) | 30.74  | 30.17  | 28.80  | 27.06  | 25.21  |
> | ViLCo (run 4) | 31.76  | 30.89  | 24.25  | 28.02  | 23.76  |
> | ViLCo (run 5) | 28.38  | 25.13  | 26.95  | 27.85  | 24.12  |
> | ViLCo (run 6) | 29.71  | 26.41  | 24.50  | 25.45  | 25.96  |
>
> From Tables 1 and 2, we observe varying performance based on the order of exposure to different data/sub-tasks. It is important to note that each Task (1-5) incorporates different action categories and sample sizes. Sub-tasks with limited data often exhibit long-tail distributions, leading to potential overfitting if these tasks are introduced early, which degrades performance in subsequent tasks. This explains why ViLCo exhibits a higher Avg. R@1 with increased BWF, highlighting that task order is a significant source of variance in continual learning. Based on your comment, we revised the paragraph in our manuscript.
>
> The visualization for random ordering ablations can be seen in the uploaded PDF.
>
> ---
> **Q2: Vision Encoders**
> >  Have the authors considered any other backbones for the visual features, such as TimeSformer, X3D, ViViT, MViT, or VideoMAE?
>
> Thank you for the valuable suggestion. In response, we expanded our experiments to include various visual backbones as the vision encoder, including Timersfomer, X3D, and ViViT. The results are presented in the table below. We believe this addition has significantly enhanced the comprehensiveness of our work and the robustness of our leaderboard.
>
> Due to time constraints, we were only able to complete experiments for the MQ task. However, we plan to include results for the NLQ and VQ tasks in the final version of the manuscript.
>
> **Table 3. Performance comparison with different visual backbones on MQ.**
>
> | Method  | Visual Backbone  | BwF | R@1 (IoU=0.3) | R@1 (IoU=0.5) | R@5 (IoU=0.3) | R@5 (IoU=0.5) |
> |---------|-----------------|-----|---------------|---------------|---------------|---------------|
> | ViLCo   | EgoVLP-V2 [1]   | 2.9 | 33.58         | 26.24         | 53.75         | 42.30         |
> | ViLCo   | Timersformer [2]| 2.4 | 30.80         | 22.82         | 51.93         | 40.64         |
> | ViLCo   | X3D [3]         | 1.4 | 31.50         | 23.01         | 48.09         | 36.59         |
> | ViLCo   | ViViT [4]       | 1.2 | 40.00         | 35.82         | 56.05         | 47.40         |
>
>
> ### Limitations:
>
> **L1: Addressing limitations of VQA and Video Captioning tasks in future works.**
> > The current limitations for tasks like VQA and video captioning are not supported by the ViLCo method. Future works address more diversity and the possibility of more sophisticated natural language; what is still unclear is how the current limitation could be investigated or addressed in the future.
>
> Our primary focus in this work is on advancing the field of continual learning, particularly in video-language tasks. While VQA and video captioning are important, they fall outside the main scope of ViLCo's objective, which is to evaluate and improve models’ ability to learn continuously from sequential data.
>
> We recognize the current limitations in these specific tasks and have updated our limitations and future work sections. We have also updated our repository with instructions and required scripts to show how others can leverage our benchmark for tasks like VQA and video captioning.
>
> Pllease visit:
> [Add New Task Documentation](https://github.com/cruiseresearchgroup/ViLCo/blob/main/Add_New_Task.md)
>
> We look forward to expanding the tasks featured on our leaderboard in future work.
>
> ---
>
> ### Related Prior Works:
> > Table 1 points out the most related benchmarks such as CLiMB although vCLIMB is the closest to the video data type.
>
> We agree that vCLIMB is more relevant as it pertains to videos. However, while vCLIMB focuses purely on video data, CLiMB addresses the intersection of language and image data through language-based queries on image data. Thus, we believe both are relevant to our work in different ways, hence we have mentioned both in the table.

---

> > ### Author Response · Authors · 2024-08-21
> > **Official Comment by Authors**
> >
> > Dear reviewer EZFs, we have taken all your comments very seriously, and have revised the paper based on your valuable comments. We sincerely hope to utilize the remaining time to engage in a productive dialogue with you. We would like to hear your thoughts given our hard work in preparing the rebuttal and additional experiments (please see the global rebuttal at the top of the page as well as the individual rebuttal for you), and the revision done. Thank you again for your kind attention.

---

### Author Rebuttal · Authors · 2024-08-17

We sincerely thank all the reviewers for the comments and feedback that have greatly improved the paper quality. We have taken all your comments seriously. We have added the new experiments, revised the writing, and improved the documentation of our repository. The following global rebuttal provides a summary of the major changes done during the rebuttal period.

## Documentation and Project's Repository

Following the suggestions from reviewer NAFV, we have extensively updated the project’s repository to include:

- More comprehensive details on how to access and build the dataset.
- Correct links to download the dataset and video features.
- Scripts to train the model for each task.

Please find the updated version [here](https://github.com/cruiseresearchgroup/ViLCo/blob/main/README.md).

Also, we added more instructions on how to add and evaluate new tasks [here](https://github.com/cruiseresearchgroup/ViLCo/blob/main/Add_New_Task.md).


## New Experiments

Following the suggestions from reviewer  EZFs, we present additional experiments in the tables below, comparing both the BWF (Backward Forgetting) and Average Recall (Avg. R@1) with an Intersection over Union (IoU) threshold of 0.5. Each run indicates a new random task ordering.


**Table 1. BwF Comparison in Different Sub-Tasks on MQ.**
| Model | Task 1 | Task 2 | Task 3 | Task 4 | Task 5 |
|--------|--------|--------|--------|--------|---------------|
| ViLCo (run 1) | N/A    | 3.0    | 2.6    | 1.7    | 2.9    |
| ViLCo (run 2) | N/A    | 0.6    | 0.8    | 4.2    | 1.9    |
| ViLCo (run 3) | N/A    | 4.2    | 2.5    | 2.7    | 5.4    |
| ViLCo (run 4) | N/A    | 2.2    | 5.7    | 3.4    | 5.7    |
| ViLCo (run 5) | N/A    | 2.6    | 1.1    | 2.2    | 4.0    |
| ViLCo (run 6) | N/A    | 3.3    | 5.2    | 4.2    | 2.5    |


**Table 2. Avg. R@1 (IoU=0.5) Comparison in Different Sub-Tasks on MQ**
| Model | Task 1 | Task 2 | Task 3 | Task 4 | Task 5 |
|--------|--------|--------|--------|--------|------------|
| ViLCo (run 1) | 32.06  | 29.15  | 26.87  | 29.30  | 26.24  |
| ViLCo (run 2) | 28.82  | 28.13  | 24.35  | 24.26  | 23.02  |
| ViLCo (run 3) | 30.74  | 30.17  | 28.80  | 27.06  | 25.21  |
| ViLCo (run 4) | 31.76  | 30.89  | 24.25  | 28.02  | 23.76  |
| ViLCo (run 5) | 28.38  | 25.13  | 26.95  | 27.85  | 24.12  |
| ViLCo (run 6) | 29.71  | 26.41  | 24.50  | 25.45  | 25.96  |

We also expanded our experiments to include various visual backbones as the vision encoder, including Timersfomer, X3D, and ViViT. The results are presented in the table below. We believe this addition has significantly enhanced the comprehensiveness of our work and the robustness of our leaderboard.

Due to time constraints, we were only able to complete experiments for the MQ task. However, we plan to include results for the NLQ and VQ tasks in the final version of the manuscript.


**Table 3. Performance comparison with different visual backbones on MQ.**

| Method  | Visual Backbone  | BwF | R@1 (IoU=0.3) | R@1 (IoU=0.5) | R@5 (IoU=0.3) | R@5 (IoU=0.5) |
|---------|-----------------|-----|---------------|---------------|---------------|---------------|
| ViLCo   | EgoVLP-V2 [1]   | 2.9 | 33.58         | 26.24         | 53.75         | 42.30         |
| ViLCo   | Timersformer [2]| 2.4 | 30.80         | 22.82         | 51.93         | 40.64         |
| ViLCo   | X3D [3]         | 1.4 | 31.50         | 23.01         | 48.09         | 36.59         |
| ViLCo   | ViViT [4]       | 1.2 | 40.00         | 35.82         | 56.05         | 47.40         |

[1] Pramanick, Shraman, et al. "Egovlpv2: Egocentric video-language pre-training with fusion in the backbone."ICCV. 2023.

[2] Bertasius, Gedas, Heng Wang, and Lorenzo Torresani. "Is space-time attention all you need for video understanding?." ICML. Vol. 2. No. 3. 2021.

[3] Feichtenhofer, Christoph. "X3d: Expanding architectures for efficient video recognition." CVPR. 2020.

[4] Arnab, Anurag, et al. "Vivit: A video vision transformer." ICCV. 2021.


Following the suggestions from reviewer NAFV, we integrated new data from MSRVTT-QA for VQA tasks. The evaluation results are shown below.


**Table 4. Performance Comparison with Different CL Methods on MSRVTT-QA**

| Method | BwF | Accuracy |
|--------|-----|----------|
| EWC    | 8.53| 17.3%    |
| iCaRL  | 3.8 | 21.6%    |
| ViLCo  | 2.4 | 24.8%    |



## Detailed responses.

We have added a rebuttal to each reviewer with more details, and we hope to discuss them with you. Thank you again for giving us your time and detailed feedback that have certainly improved the quality of our paper.

---

### Comment · Area_Chair_qsj7 · 2024-08-21

Dear Reviewers,

The author has responded to the comments and concerns raised by the reviewers. We kindly ask that you review the author’s responses and provide feedback at your earliest convenience.

Bests,
AC

---

### Comment · Area_Chair_qsj7 · 2024-08-30

Dear Reviewers,

The Discussion Period is ending soon. Reviewers who haven’t responded to the author’s rebuttal yet, please do so as soon as possible. Other reviewers may also use the remaining time to further discuss with the author. Thank you for your efforts.

Bests,
AC

---

### Decision · Program_Chairs · 2024-09-26

**Decision:**

Accept (Poster)

**Comment:**

The paper introduces ViLCo-Bench, a new benchmark for video-language continual learning, which is praised for its novel approach to addressing the challenges of combining video and language tasks. Strengths include the creation of a standardized framework for evaluating continual learning models, incorporating self-supervised learning and addressing memory efficiency issues. Reviewers appreciate the empirical results showing potential benefits in mitigating catastrophic forgetting. However, concerns are raised about the limited diversity of the dataset, complexity in writing, and missing citations. Despite these issues, the benchmark is considered a valuable contribution. Hence, I recommend to accept this paper.